# Quantitative Proteomics of Medium-Sized Extracellular Vesicle-Enriched Plasma of Lacunar Infarction for the Discovery of Prognostic Biomarkers

**DOI:** 10.3390/ijms231911670

**Published:** 2022-10-01

**Authors:** Arnab Datta, Christopher Chen, Yong-Gui Gao, Siu Kwan Sze

**Affiliations:** 1Laboratory of Translational Neuroscience, Division of Neuroscience, Yenepoya Research Center, Yenepoya (Deemed to be University), University Road, Deralakatte, Mangalore 575018, Karnataka, India; 2School of Biological Sciences, Nanyang Technological University, 60 Nanyang Drive, Singapore 637551, Singapore; 3Memory, Aging and Cognition Centre, National University Health System, Singapore 119228, Singapore; 4Department of Pharmacology, Yong Loo Lin School of Medicine, National University of Singapore, Singapore 119077, Singapore; 5Department of Health Sciences, Faculty of Applied Health Sciences, Brock University, 1812 Sir Isaac Brock Way, St. Catharines, ON L2S 3A1, Canada

**Keywords:** lacunar stroke, prognostic biomarker, plasma biomarker, extracellular vesicles, iTRAQ, mass spectrometry

## Abstract

Lacunar infarction (LACI), a subtype of acute ischemic stroke, has poor mid- to long-term prognosis due to recurrent vascular events or incident dementia which is difficult to predict using existing clinical data. Herein, we aim to discover blood-based biomarkers for LACI as a complementary prognostic tool. Convalescent plasma was collected from forty-five patients following a non-disabling LACI along with seventeen matched control subjects. The patients were followed up prospectively for up to five years to record an occurrence of adverse outcome and grouped accordingly (i.e., LACI-no adverse outcome, LACI-recurrent vascular event, and LACI-cognitive decline without any recurrence of vascular events). Medium-sized extracellular vesicles (MEVs), isolated from the pooled plasma of four groups, were analyzed by stable isotope labeling and 2D-LC-MS/MS. Out of 573 (FDR < 1%) quantified proteins, 146 showed significant changes in at least one LACI group when compared to matched healthy control. A systems analysis revealed that major elements (~85%) of the MEV proteome are different from the proteome of small-sized extracellular vesicles obtained from the same pooled plasma. The altered MEV proteins in LACI patients are mostly reduced in abundance. The majority of the shortlisted MEV proteins are not linked to commonly studied biological processes such as coagulation, fibrinolysis, or inflammation. Instead, they are linked to oxygen-glucose deprivation, endo-lysosomal trafficking, glucose transport, and iron homeostasis. The dataset is provided as a web-based data resource to facilitate meta-analysis, data integration, and targeted large-scale validation.

## 1. Introduction

Lacunar infarction (LACI) or lacunar stroke is a sub-type of ischemic stroke where tissues supplied by small cerebral blood vessels are infarcted [1]. It constitutes up to a quarter of all ischemic stroke cases. LACI has an increased prevalence among people originating from the Indian subcontinent (India, Sri Lanka, Pakistan, Nepal, and Bangladesh) when compared to whites [2] and is etiologically the most common subtype in the Japanese population [3]. Despite a high prevalence among Asian populations, LACI remains poorly understood, misdiagnosed, and underdiagnosed compared to non-lacunar strokes. The gradual progression of neurological symptoms and motor deficits during the initial stage is more common after LACI compared to non-lacunar infarcts [4]. Despite favorable short-term prognosis due to higher survival than other ischemic stroke subtypes, LACI increases the mid- to long-term risk of developing recurrent vascular events or dementia [5]. In the United States alone, up to half of ischemic stroke patients who are on anti-platelet therapy develop a recurrent ischemic stroke [6]. Hence, a well-defined protocol for accurate mid- to long-term prognosis is essential to segregate patients at risk of vascular complications or adverse functional outcome. Blood biomarkers can act as a complementary tool to neuroimaging and clinical examination to predict prognosis in LACI patients.

The majority of blood-based biomarker studies for LACI or cerebral small vessel disease (cSVD) have dealt with only a handful of candidate proteins selected through any one or a combination of the following criteria: (1) close association with biological processes such as coagulation, fibrinolytic cascade (e.g., fibrinogen), endothelial dysfunction, and inflammation; (2) highly abundant components of plasma proteome (e.g., alpha-2-macroglobulin, albumin, apolipoprotein B, apolipoprotein A, lipoprotein A); (3) already in clinical use for other vascular pathologies such as cardiovascular diseases (e.g., C-reactive protein (CRP), NT-proBNP [7]) or blood coagulation disorders (e.g., D-dimer); and (4) easy availability of robust antibodies for performing protein arrays or ELISA or flow cytometry [8,9,10,11]. An inevitable outcome of this reductionist strategy of focusing on a very small subset of plasma proteome is proposing the same protein as a biomarker for similar vascular diseases that may compromise the specificity of the candidate. Until now, no clinically approved biomarkers exist for the prognosis of LACI or for stroke as a whole.

A majority of biomarker studies on cSVD or LACI have focused on plasma/serum samples that were collected during the acute phase (within hours to days), whereas for prognosis, mostly short-term outcomes (within a few months) were assessed [8]. LACI is known to be associated with a chronic, subtle, and diffuse blood–brain barrier (BBB) dysfunction in the white matter [12]. Several pre-clinical and clinical studies have indicated that different cell types of brain parenchyma and vascular endothelial cells can secrete extracellular vesicles (EVs) following ischemic stress or in response to oxygen-glucose deprivation (OGD) [13]. Owing to this partially compromised BBB, a part of these secreted EVs can enter the general circulation that may be associated with the underlying neurovascular pathologies [14]. Hence, selectively targeting blood EVs could be a novel approach, especially for the discovery of prognostic biomarkers of LACI that precedes a leakage of BBB. Further, the use of EVs can partially circumvent the problem of extreme dynamic range and masking effect of high-abundance plasma proteins associated with whole plasma proteomics profiling studies. Until now, the shotgun proteomics studies that aimed to discover biomarkers for future ischemic stroke from serum EVs [15] or for large vessel disease from whole plasma [16] or for LACI from plasma high-density lipoprotein particles [17] detected a small number of proteins (n ≤ 55) due to masking of the low abundant portion of the proteome by high-abundance plasma proteins during protein mass spectrometry.

Our group was the first to report the differential proteome of small-extracellular vesicles (SEV) (200,000 g, exosome-rich fraction) from plasma samples of LACI patients to predict prognosis [18,19] that were obtained from a registered randomized clinical trial on stroke [20,21,22]. In that study, the pooled plasma samples were depleted of medium-sized extracellular vesicles (MEVs) (mentioned as plasma membrane-derived vesicles, or microparticles in refs. [18,19]) by sequential centrifugation and ultracentrifugation. Here, we report the quantitative proteomics profile of the MEVs that was obtained from the same pooled plasma and was processed using an identical iTRAQ protocol [18]. This dataset allowed us (a) to propose a shortlist of MEV proteins for individual validation on an independent cohort of LACI patients, a majority of which are novel, unique from the SEV proteome, and are not linked to commonly studied biological processes in the context of LACI; and (b) to create a publicly accessible web resource on the MEV proteome useful for future studies on LACI, related neurological disorders, or on EV-based biomarker discovery.

## 2. Results

Quantitative protein mass spectrometry using 4-plex iTRAQ on MEV-enriched plasma samples identified 573 proteins with at least one peptide having a confidence >95% (unused protein score ≥ 1.3, peptide FDR < 1%). The complete dataset is provided to users at https://yenepoya.res.in/database/LTN_Datta_Lab/LACI_MEV_Proteomics/index.html (accessed on 19 August 2022). The mass spectrometry-generated raw files have been deposited to the ProteomeXchange Consortium via the PRIDE partner repository with the dataset identifier PXD032225 (Figure 1).

### 2.1. Quality Control of Plasma-Derived MEV

Plasma-derived MEVs (also called plasma membrane-derived vesicles, microvesicles, or microparticles) can be contaminated with soluble proteins such as lipoproteins and SEVs such as exosomes. To characterize the MEVs based on the protein composition, we followed the MISEV2018 guidelines [23]. There are five categories of EV markers where two categories (categories 1 and 2) are generally considered as positive controls for all EVs while category 3 markers are considered as common contaminants. Several quantified proteins in our dataset such as integrins, heterotrimeric G proteins, transferrin receptor, and MHC class I proteins that are non-tissue specific belong to category 1a EV markers. Few cell/tissue-specific EV markers such as integrin alpha-IIb (ITGA2B or CD41) (platelet), platelet endothelial cell adhesion molecule (PECAM1) (endothelial cells), glycophorin (RBC) from category 1b were quantified. Apolipoprotein B-100 (APOB), serum albumin (ALB), apolipoprotein A-I (APOA1), and apolipoprotein A-II (APOA2) were present as common contaminants (category 3) that were co-isolated with EVs as expected when a separation approach of intermediate specificity (i.e., differential centrifugation/ultracentrifugation) is used during the isolation of MEVs. The markers of category 4 that are originated from subcellular structures such as the nucleus, mitochondria, endoplasmic reticulum, or Golgi apparatus are generally found in large EVs (such as MEVs) and are not enriched a priori in SEVs [23]. Indeed, we detected at least nine proteins (e.g., histone H2A, endoplasmin, isoform 2 of calnexin) from category 4 in our MEV dataset, eight of which were absent in the SEV (exosome-rich fraction) dataset [18] that were obtained from the same plasma sample and processed at the same time. Few EV markers from category 5 that includes cytokines or growth factors such as transforming growth factor beta-1 (TGFB1) were also detected. Overall, of 573 identified proteins, 53 proteins were detected from categories 1, 2, 4, and 5 (Appendix A). Of 53 proteins, 14 were detected in common between MEV and SEV fractions (Appendix A). This contains only one protein (histone H4) from category 4 that is also present in the SEV fraction. Overall, our results confirm that (1) the MEV samples are largely free of SEV contamination and (2) the isolated MEVs are of intermediate purity as per the MISEV2018 guidelines.

### 2.2. Adverse Outcome Is Associated with an Increase in the Total Number of Altered Proteins

To identify the general pattern of protein abundance changes in LACI patients with favorable and adverse outcome compared to healthy control (HC), frequency histograms of log2 ratios (Appendix A) and volcano plots were drawn (Figure 2A–C).

Of the 573 quantified proteins, 69 (12%), 122 (21%) and 112 (20%) proteins were classified as altered in ‘no adverse outcome’ (NAO), ‘recurrent vascular event’ (RVE) and ‘cognitive decline’ (CD) groups, respectively, based on the dual criteria: |Log_2_(ratio)| > 0.58 and (−log_10_(*p*-value)) > 1.30. This is equivalent to a fold change of 1.5-fold and a *p*-value of 0.05. Overall, 146 (25%) proteins were shortlisted. Each of these proteins is altered in at least one of the three LACI groups when compared to HC. This list can be accessed at https://yenepoya.res.in/database/LTN_Datta_Lab/LACI_MEV_Proteomics/Altered_Protein.html (accessed on 19 August 2022). The total number of altered proteins was much higher for RVE or CD groups when compared to the NAO group. The trend remained consistent irrespective of the *p*-value (e.g., 0.025, 0.01, 0.005) used for filtering the complete dataset while the trend become clearer with an increase in the magnitude of change (e.g., fold change: 2, 2.5) (Appendix A).

Next, to specify the effect of LACI on the plasma MEV proteome, we counted the number of increased and decreased proteins in each of the three groups. In NAO, RVE, and CD groups, 52 (out of 69, 74%), 109 (out of 122, 89%), and 87 (out of 112, 78%) proteins were decreased, respectively, in abundance (Figure 2D). Therefore, major elements of the perturbed proteome are decreased following LACI irrespective of the grouping. Further, the proportion of decreased proteins appeared to be higher in LACI groups with adverse outcome (Figure 2D) compared to the NAO group. This points toward an underlying link with the pathology of LACI. Together, these results suggest that: (1) adverse outcome in LACI patients is associated with an increase in the total number of altered plasma proteins; (2) altered proteome in LACI patients is contributed predominantly by proteins that are reduced in abundance.

Figure 3 presents a Venn diagram of the altered proteome (n = 146) showing overlapping and group-specific protein counts among the three LACI groups. Based on this, the altered proteome can be categorized into three classes; (1) LACI-related: proteins that are altered in all three LACI groups (51/573, 9%) irrespective of the outcome; (2) outcome-dependent: proteins that are commonly altered in the adverse outcome (RVE and CD) groups (43/573, 8%); (3) group-specific: proteins that are altered in each of the three LACI groups specifically (NAO, 6/146; RVE, 18/146; CD, 16/146). Similarly, outcome-related and group-specific alterations of RVE and CD groups together (i.e., 43 + 18 + 16 = 77 proteins, 13%) could be useful for predicting an adverse outcome. Overall, these results suggest that 1) the majority of quantified plasma proteins (75%, 427/573) remain unchanged following LACI, and 2) adverse outcome in LACI patients is linked to relatively small and specific changes in the plasma MEV proteome (13%, 77/573).

### 2.3. Proteome-Wide Correlation Analysis Reveals Differing Patterns Depending on Outcome

To compare the overall trends of protein alteration in different LACI groups compared to the HC group, we performed group-wise bivariate correlation analysis using Spearman rank correlation. The protein ratios in between NAO and the adverse outcome groups showed moderate (ρ = 0.60, NAO vs RVE, *p* < 0.01; ρ = 0.63, NAO vs CD, *p* < 0.01) positive correlations when complete proteome (n = 573) was used for analysis **(Figure 4**A, B). In contrast, for a similar analysis between RVE and CD groups, a strong (ρ = 0.86, *p* < 0.01, RVE vs CD) positive correlation was observed (Figure 4C). No major difference in correlation coefficients was seen if we select various subgroups for analysis such as the altered proteome (Figure 3, n = 146), LACI-related (Figure 3, n = 51), or the outcome-dependent (Figure 3, n = 43) proteome (Appendix A). Overall, at baseline, the general pattern of the proteomic landscape of the two adverse outcome groups (RVE and CD) was different compared to the NAO group.

Next, we investigated how distinct were the patterns of alteration of the group-specific proteome by selecting group-specific altered proteins (Figure 3) from the three LACI groups and performing Spearman rank correlation analysis between these three groups (Figure 5). The abundance changes in proteins altered only in NAO showed either weak positive (ρ = 0.31) or moderate negative (ρ = −0.60) correlation (Figure 5A,D) with RVE and CD groups, respectively. The NAO group consists of 3 proteins with increased abundance (e.g., apolipoprotein(a) (LPA), serum paraoxonase/arylesterase 1 (PON1), catalase (CAT)) and 3 proteins with decreased abundance (e.g., transitional endoplasmic reticulum ATPase (VCP), complement C1r subcomponent (C1R), transferrin receptor protein 1 (TFRC)). Presumably, these proteins would be candidates for future studies to predict positive outcome following an event of LACI. The RVE-specific changes were moderately correlated with NAO (ρ = 0.47) and weakly correlated with CD (ρ = 0.26) group (Figure 5B,E). For CD-specific proteins, no correlation (ρ = 0.10) was observed when compared to NAO, while a strong positive correlation (ρ = 0.89, *p* < 0.01) was seen with the RVE group (Figure 5C,F). The CD group contains elevated proteins such as haptoglobin-related protein (HPR), pregnancy zone protein (PZP), or decreased proteins such as NAD(P) transhydrogenase, mitochondrial (NNT). Some of these proteins could be useful in differentiating CD from NAO but not from RVE.

### 2.4. Convalescent Plasma MEV—Preferred Fraction for Biomarker Discovery

We purposefully collected the plasma of LACI patients with a median delay of 47 days to allow the acute systemic immunological and inflammatory response to subside. We believed this will help us to discover novel biomarkers beyond the small subset of extensively studied candidates, a part of which is predominantly linked to inflammation. We identified inflammation/immune-response proteins as those with Gene Ontology (GO) Biological Process terms containing the strings “inflammation”, “inflammatory”, “immune”, and “immuno”. Of 146 altered proteins, 28 (19%) were associated with these terms. Of 423 unaltered proteins, 68 (16%) were associated (χ^2^, *p* > 0.05). Hence, inflammation-related proteins were not enriched in the altered MEV proteome. In contrast, the deregulated MEV proteome was found to be preferentially associated with various types of vesicles (GO Cellular Component terms: “vesicle”, “endosome”, “lysosome”, “autophagosome” and “secretory granule”; 43/146, 36%; 96/423, 23%; χ^2^, *p* = 0.001) (Appendix A). A part of the vesicle-associated proteins is known to be secreted and is more likely to reach general circulation through a leaky BBB in LACI patients [12]. Overall, targeting MEV from the convalescent plasma helped to partially evade the acute inflammatory response while selectively concentrating vesicular or secreted proteins in the perturbed component of the MEV proteome.

### 2.5. Plasma MEV Contain Disease-Specific Signatures of Key Pathological Events

Coagulation or clotting and fibrinolysis remain the most widely targeted biological process for the validation of plasma biomarkers of LACI or cSVD in human cohorts in population-based or hospital-based studies [9]. Using GO Biological Process, we identified coagulation-associated proteins (GO term: coagulation, fibrin, clot, fibrinolysis, aggregation) in deregulated (34 out of 146, 23%) and unchanged (40 out of 423, 9%) plasma MEV proteome. In line with the above notion, the results show highly significant association of coagulation-related proteins in the deregulated MEV proteome using a chi-square analysis (χ^2^, *p* < 0.001) (Appendix A).

LACI is a vasculature-linked disorder that occurs due to the thickening of the vascular wall and subsequent narrowing of deep penetrating cerebral arteries [24]. This causes endothelial dysfunction and an impaired autoregulation of cerebral blood flow [9]. Silent or symptomatic LACI can also be associated with systemic endothelial dysfunction that is present in vascular beds of far-off organs such as the kidney [25] or brachial artery [26]. To test if this pathological attribute of generalized endothelial dysfunction is reflected in the plasma MEV proteome, we identified the vascular endothelium-associated proteins using GO Biological Process (GO term: vasculature, vascular, endothelial). Twenty-one out of 146 deregulated proteins (14%) were vasculature-associated, while 31 of out 423 unchanged proteins (7%) were vasculature-associated (χ^2^, *p* = 0.011). A similar analysis using GO Cellular Component to find out the enrichment of brain-associated proteins among altered proteome did not give significant association by chi-square test (χ^2^, *p* > 0.05) (Appendix A). Considering the plasma MEV proteome is contributed by proteins from all organs, this result is not surprising and, in a way, shows the specificity of our enrichment analysis.

OGD and/or hypoxia of brain tissue constitute the central pathological event that is common to all subtypes of ischemic stroke [14,27]. To test if plasma MEV of LACI patients carries a signature of this cellular event that is specific to the pathology of interest, we identified proteins that are linked to biological processes that contain “OGD” and “hypoxia” using GO Biological Process. Thirteen out of 146 deregulated proteins (9%) were linked to OGD/hypoxia compared to 13 out of 423 (4%) proteins that did not show any change in the plasma levels following LACI (χ^2^, *p* = 0.004). Of note, the average age of the study cohort was more than 60 years. To check the disease specificity of this analysis, we performed a similar analysis on amyloid-associated biological processes on deregulated (7/146, 5%) and unchanged (8/423, 2%) proteomes. This time no significant enrichment was observed by chi-square analysis (χ^2^, *p* > 0.05) (Appendix A).

Overall, GO coupled with an unbiased enrichment analysis indicated that (1) LACI causes specific alteration of vesicle-, coagulation-associated, and vascular endothelium-linked proteins in the MEV proteome, and (2) proteins associated with pathological events that are seen at the cellular level such as OGD or hypoxia are also enriched in the altered MEV proteome.

### 2.6. Majority of Adverse Outcome Predictors Are Not Linked to Coagulation or Inflammation

One of the primary objectives of this discovery study is to propose a list of candidate prognostic biomarkers for large-scale validation. We filtered the altered list of 146 proteins based on the following criteria: (1) The proteins that are altered in all three LACI groups with a uniform direction of regulation are disfavored. A majority of LACI-induced proteins (45/51, 88%) showing similar regulation fell into this category. (2) The proteins that are altered exclusively in only the NAO group or in either one or both adverse outcome groups are favored. E.g., VCP, moesin (MSN), cholinesterase (BCHE). (3) Proteins that are associated with relevant GO terms such as OGD, hypoxia, oxidative stress, vesicle, or vasculature are preferred. (4) Proteins that showed an opposite regulation between NAO and either one or both adverse outcome groups are preferred. E.g., IgGFc-binding protein (FCGBP), galectin-3-binding protein (LGALS3BP), and isoform 15 of fibronectin (FN1) are decreased in abundance in NAO while showing an increase in both RVE and CD groups. Conversely, band 3 anion transport protein (SLC4A1) is increased in abundance in NAO and is decreased in RVE and CD groups. (5) Irrespective of the *p*-value, proteins (e.g., AHNAK, ROCK2, C5, CP) that showed a similar direction and magnitude of change between NAO and RVE/CD were not included. This resulted in a shortlist of 63 proteins (Table 1). It can be accessed and downloaded from the data resource: https://yenepoya.res.in/database/LTN_Datta_Lab/LACI_MEV_Proteomics/LACI_Biomarker_Shortlist.html (accessed on 19 August 2022).

Of note, a majority (34/63, 54%) of these candidates are not linked to inflammation, complement activation, coagulation, fibrinolysis, or endothelium as per GO-term analysis. To the best of our knowledge, several proteins in Table 1 are novel and have not been reported as potential biomarkers in the context of LACI or non-lacunar sub-types of strokes.

### 2.7. MEV-Proteome Is Qualitatively and Quantitatively Different from SEV Proteome

Availability of the SEV proteomics dataset from the same cohort of patients and source plasma [18] provided us with a unique opportunity to compare the proteome-wide presence and relative abundance of individual proteins between MEV and SEV datasets in various LACI groups. Ninety-one proteins were identified in common between MEV (n = 573) and SEV (n = 344) datasets. About 14.5% of quantified MEV proteins (83/573) were also quantified in the SEV dataset. To visualize the relative distribution of individual proteins among differentially centrifuged fractions, correlation plots of protein ratios of commonly quantified 83 proteins between MEV and SEV datasets for different LACI outcome groups were drawn (Figure 6). For the NAO group, weak positive correlation (ρ = 0.26, MEV vs SEV, *p* < 0.05) was observed while for the RVE group, weak negative correlation (ρ = −0.27, MEV vs SEV, *p* < 0.05) was observed. No correlation was seen in the CD group among MEV and SEV datasets. Overall, this analysis shows that (1) when identical techniques and parameters are used for isolation, acquisition, and analysis, MEV provides better proteomic coverage compared to SEV; (2) major elements of the MEV proteome (~85%) are different from the SEV proteome; (3) among the common elements (14.5%), the relative distribution of individual proteins’ abundances is not uniform between moderate (i.e., MEV) and high-speed (i.e., SEV) centrifuged fractions for the NAO group which worsens in the adverse outcome (CD, no correlation; RVE, weak negative correlation) groups.

## 3. Discussion

The proteomic characterization of MEV was performed using plasma samples of LACI patients who participated in the ESPRIT-cog sub-study [22] that was a part of a registered randomized interventional clinical trial, ESPRIT [20,21]. MEVs are biochemically, morphologically, and biophysically distinct from SEVs (e.g., exosomes) due to their distinct protein composition. The abundance of one protein in the MEV fraction does not predict its abundance in the SEV fraction or vice versa (Figure 6). The depth of proteomics coverage is higher for MEVs (n = 573) than SEVs (n = 344) when both types of EVs are isolated in parallel from the same starting sample, processed in parallel using an identical methodology, and identical filters are used for data analysis. Unlike SEVs, MEVs (or microparticles) are relatively easier and faster to isolate making them an attractive starting material for circulatory biomarker discovery.

No universally accepted method of isolation is available to obtain different EVs in pure form. It is accepted that MEV sediments with a gravitational force between 10,000 to 30,000 g [28]. Hence, it does not necessarily need an ultracentrifugation step for isolation. In our study, due to a low volume of starting sample, the protein yield was insufficient from either the P2 (12,000 g fraction) or P3 (30,000 g fraction) fraction individually. We combined these two fractions to obtain sufficient quantities of MEVs for the proteomics analysis. As recommended by MISEV2018 guidelines, the presence of several MEV-specific markers and depletion of most of these markers in the SEV fraction indicated that the MEV samples are of intermediate purity and specificity (Appendix A) [23]. Of note, no EV separation method has been developed that can achieve high recovery with high specificity simultaneously.

A majority of the studies that targeted EVs of ischemic stroke for biomarker discovery sampled blood during the acute phase (hours to a few days) of stroke [29]. The onset of an acute systemic inflammatory response apart from rapid temporal changes in the pathological state of the neurovascular unit of the affected brain tissue during the initial few days can increase intragroup variation. This may confound data interpretation and may decrease the specificity of the potential biomarkers. We attempted to bypass this through a dual strategy of collecting the convalescent plasma samples while disfavoring inflammation-related proteins during data filtering. The absence of enrichment of immune response, complement system, or inflammation-related proteins in the deregulated proteome indicates the success of the above strategy.

For biomarker discovery from plasma samples of stroke patients, individual variation, arising from unaccounted factors, is one of the major confounders. Even in healthy donors, the protein composition of plasma-derived microvesicles varied widely between individuals as seen in proteomics analysis [30]. To overcome individual variation, a group-wise sample pooling strategy was adopted. Sample pooling has been used widely for clinical proteomics studies by us and others as an effective strategy to identify changes that are real [16,19].

LACI causes a specific alteration of (146/573, 25%) one-quarter of all quantified proteins. This is reflected in the preferential association of the deregulated proteome to key cellular events that trigger the disorder such as OGD, hypoxia, and relevant anatomical structures that are the sites of initiation of the disorder such as vasculature or endothelium. As patients from all three LACI groups suffered from a single event of a lacunar stroke, protein ratios were broadly similar among all LACI patients (Figure 3). The patients from the NAO group did not report any adverse outcome while the other two groups had adverse outcome when followed up prospectively for up to 5 yrs. This is reflected in part (1) by a baseline increase in the total number of altered proteins in the adverse outcome groups (RVE, CD) compared to the NAO group (Figure 2), and (2) by a strong proteome-wide correlation between RVE and CD groups (Figure 4C and Figure 5F). Of note, although none of the patients in the CD group had any recurrent vascular event, more than half of the patients in the RVE group (6 out of 11) did have incident cognitive decline. This explains why the MEV proteome of RVE and CD groups is largely similar. Presumably, the CD group remained asymptomatic in terms of vascular complications, unlike the RVE group.

One interesting observation in this study is that LACI causes a general reduction in the abundance of a majority of MEV proteins when compared to the HC group (Figure 2D). Circulatory EV concentration is known to increase during acute (within 24 h) [31] and sub-acute phases (median sampling time: 2.8 days) [32] of ischemic stroke when compared to non-stroke controls. In contrast, during the chronic stage (after 6 months), circulatory EVs (e.g., platelet-derived microparticles) are reduced in a cohort of cSVD patients [33]. This corroborates our observation. Further, in normal platelets in vitro, aspirin inhibits the arachidonic acid-induced platelet reactivity, EV formation, and procoagulant activity [34]. Hence, considering all LACI patients were treated with aspirin, this may further contribute to this generalized reduction in the MEV proteome when compared to the HC group.

Despite being on treatment with antiplatelet/antithrombotic drugs, a proportion of LACI patients suffered from recurrent vascular events. A subset of the differentially regulated proteins between NAO and adverse outcome groups may partly explain this therapeutic failure. E.g., two proteins with known antioxidative properties (PON1, CAT) were elevated selectively in the NAO group. PON1, a high-density lipoprotein (HDL)-associated organophosphate ester hydrolase, is atheroprotective due to its strong antioxidant properties [35]. SLC2A1 (GLUT1) is the major glucose transporter in the mammalian blood–brain barrier and is expressed on the cell surface of brain vascular endothelial cells. It is reduced in the RVE group while showing an increasing trend in the NAO group. In mild Alzheimer’s disease patients, GLUT1 expression was also reduced in the brain-derived circulating endothelial cells that were isolated from peripheral blood samples when compared to healthy control [36]. Hemopexin (HPX) and serotransferrin (TF) that are increased in the NAO group and decreased in the RVE group are responsible for scavenging proinflammatory and prooxidant heme and for binding and transporting free iron, respectively. Both HPX and TF were also increased in the affected hemisphere of rats following middle cerebral artery occlusion and subsequent reperfusion [37]. The reduction in clathrin coat adaptor protein, AP2B1 (isoform 2 of AP-2 complex subunit beta) in the RVE group indicates that defective intracellular trafficking and endo-lysosomal processes may contribute to post-LACI vascular complications.

There are candidates from the shortlist that have been proposed as potential biomarkers for different types of vascular complications. E.g., serum VCP level was higher in patients with acute coronary syndrome when compared to healthy control or chronic coronary syndrome [38]. Serum VCP level was increased in myocardial ischemia-induced sudden cardiac deaths when compared to matched control patients who died from non-cardiac causes [39]. These results corroborate with the current study where plasma VCP level is decreased in LACI patients with NAO. Some of the above-mentioned candidates can be used to generate a multi-marker panel to improve the sensitivity and specificity of the prediction.

There are a few limitations of this discovery study. (1) The long storage duration (1999–2011; up to 12 yrs., −80 °C) of the samples should be taken into account before comparing the results of this study with similar studies during meta-analysis. Of note, being a discovery study for prognostic biomarkers, the follow-up period constitutes a major part of the waiting time. The literature on the long-term stability of circulatory proteins at ultra-low temperature (≤ 70 °C) remains limited, variable, and inconclusive as some reported no change for certain proteins (e.g., TGFB1 [40], CRP [41], insulin-like growth factor 1 [42]) or decline, while others reported a significant increase for certain proteins such as TGFB1 [42]. (2) The three LACI groups were not matched by sex and smoking. In any case, the effect of sex on the levels of biomarkers cannot be ascertained as male and female samples were pooled together. Certain blood lipoproteins such as high-density lipoprotein cholesterol (HDL-C), apolipoprotein A-1 were inversely associated with the severity of white matter lesions in women, but not in men [43]. Sexual dimorphism has been observed when blood RNA expression was compared between male and female lacunar stroke patients using a whole transcriptome array. E.g., isoform 3 of alpha-adducin (ADD1), which is decreased significantly in the RVE group, was decreased in males in the blood RNA, but increased in the case of female patients [44]. This indicates that sex could be a possible confounding factor that may increase false-negative findings. (3) A low sample amount precluded the use of single vesicle-based imaging or particle analysis techniques such as nanoparticle tracking analysis for testing the purity of the MEVs. The purity of the isolated EVs and determining the specificity of the EV subtype among various types of EVs remains an ongoing challenge. (4) The limited availability of plasma and low yield of MEVs did not allow WB/ELISA validation of any of the discovered candidates on individual LACI patients. (5) Our dataset does not provide any information on whether deregulation of a part of the aberrantly regulated EV proteome precedes the index event of LACI (i.e., potential cause of LACI) or only succeeds it (i.e., effect of LACI). Being a single-timepoint snapshot, we are unable to confirm the causality between LACI and individual biomarkers. (6) We are also unaware if these changes are predominantly palliative or restorative in nature or adversely contribute to the injury during prognosis or are just bystanders without any direct consequence to LACI pathology. A longitudinal profiling involving a time series may provide important inputs on specific candidate biomarkers.

## 4. Materials and Methods

### 4.1. Reagents

Unless indicated, all reagents were purchased from Sigma-Aldrich (St. Louis, MO, USA).

### 4.2. Sample Collection and Patient Information

The patients were recruited at the Singapore General Hospital between 1999 and 2005 into the cognitive sub-study of the European Australasian Stroke Prevention in Reversible Ischemia Trial (ESPRIT) [20,22]. The plasma samples were obtained from patients within 6 months of non-disabling ischemic stroke (grade ≤ 3 on the modified Rankin scale (mRS)) of presumed arterial origin. The plasma sample collection containing inclusion and exclusion criteria, clinical information, neuropsychological test battery, and baseline risk factors were identical as reported earlier [18]. The control plasma was collected from non-stroke subjects at the same site during 2004–2006. For the reader’s convenience, we provide the details of sample collection and patient information in the Appendix A Methods section.

### 4.3. Experimental Design Guided by Outcome Measures

The experimental design is identical as reported earlier [18,19] except for the fact that MEVs were isolated instead of SEVs from the same pooled plasma samples (Figure 1) (SI Methods). Briefly, all LACI patients were followed up annually for up to 5 years (median follow-up, 3 years; interquartile range, 2 years) to monitor for the occurrence of any vascular event or for change in the baseline cognitive status. Any LACI patient having a recurrence of vascular event during the follow-up period was included in the group called “recurrent vascular event” (RVE). The patients whose cognitive status declined from the respective baseline status during the course of the prospective study had been assigned to the “cognitive decline” (CD) group. Patients who did not suffer an RVE or CD during this period were grouped as “LACI, no adverse outcome” (NAO). Accordingly, plasma samples of 45 LACI patients were divided into three groups based on the outcome variables (LACI—NAO, n = 19; LACI—RVE, n = 11; LACI—CD but no RVE, n = 15). The control group (healthy control, HC) had 17 subjects who never had a stroke or cancer and were cognitively normal at the baseline. The demographic characteristics, delay to blood draw, baseline risk factors, and cognitive status of the study population stratified by outcome measures and HC group are summarized in Appendix A. No significant difference was observed between the three groups of LACI patients in terms of any of these variables except ‘gender’ (H(2) = 11.86, *p* = 0.003) and ‘smoking’ (H(2) = 7.276, *p* = 0.026). The plasma samples from four groups were pooled group-wise to obtain four sets of pooled plasma for proteomics processing.

### 4.4. Proteomics Sample Preparation

#### 4.4.1. Isolation of MEV-Enriched Fraction by Sequential Centrifugation and Ultracentrifugation

The whole procedure was performed at 4 °C. There were four tubes of pooled plasma from four groups each containing around 5 mL of plasma specimens. The samples were subjected to sequential centrifugation and ultracentrifugation to fractionate extracellular vesicles using a modified protocol as described previously [18,19]. Briefly, sonicated plasma (5 ×1 min) was centrifuged at 4000× *g* twice for 30 min and then at 12,000× *g* for 30 min to collect the pellets (4000× *g*—P1 and 12,000× *g*—P2). The resulting supernatant was diluted five times with 1X PBS and ultra-centrifugated (swinging bucket rotor: SW 55 Ti, k-factor = 48) at 30,000× *g* for 2 h to collect the pellet (P3). The supernatant was used for the isolation of the SEV-rich (exosome) fraction as reported earlier [18]. P3 was washed with 1X PBS twice and lyophilized. P2 and P3 were dissolved using 50 µL of ice-cold dissolution buffer (6% sodium dodecyl sulfate; 20 mM dithiothreitol, 100 mM tris-HCl with Complete Protease Inhibitor Cocktail (COMPLETE, (Roche; Mannheim, Germany)), pH 7.75) by brief vertexing. Protein quantization was performed using the 2-D Quant Kit (Amersham Biosciences, Piscataway, NJ, USA). P2 and P3 were combined to obtain MEVs for proteomics sample preparation.

#### 4.4.2. In-Gel Tryptic Digestion and Isobaric Labeling

The samples (83 μg/condition) were subjected to denaturing PAGE using a 4%–6%–25% gel following an identical procedure as described previously [19,37,45]. Briefly, the diced gel bands were extensively washed, reduced, alkylated, and digested overnight (12.5 ng/μL of sequencing-grade modified trypsin ((Promega, Madison, WI, USA), in 50 mM TEAB, 2% acetonitrile (ACN), trypsin: protein ratio: 1:20) at 37 °C. Subsequently, the peptides were extracted and dried before being labeled with respective isobaric tags of the 4-plex iTRAQ Reagent Multi-Plex Kit (Applied Biosystems, Foster City, CA, USA) as follows: HC, 114; LACI—NAO, 115; LACI—RVE, 116; LACI—CD, 117 (Figure 1). The labels were combined, desalted, and dried.

#### 4.4.3. Offline ERLIC and LC-MS/MS

A modified electrostatic repulsion and hydrophilic interaction chromatography (ERLIC) with volatile salt-containing buffers was used to fractionate the iTRAQ-labeled peptides offline. The chromatography buffer, HPLC gradient, collection of eluted fractions, and subsequent pooling were performed as described earlier [18] (SI Methods). The vacuum-dried peptides were reconstituted with 0.1% FA, 3% ACN and analyzed using an HPLC system (Shimadzu, Kyoto, Japan) coupled with QSTAR Elite Hybrid MS (Applied Biosystems/MDS-SCIEX) as described previously with minor modifications [19]. The samples were injected thrice during LC-MS/MS analysis (technical replicate = 3). Briefly, most of the LC parameters for a 90 min gradient including column configuration, gradient, and flow rate were kept constant except the mobile phase A composition (0.1% FA in 3% ACN) and sample injection volume (15 µL/injection). Regarding MS parameters, the precursors with a mass range of 300–1600 m/z and calculated charge of + 2 to + 5 were selected for the fragmentation. The selected precursor ion was dynamically excluded for 20 s with a 50 mDa mass tolerance. The maximum accumulation time was set at 1.0 s. All other MS parameters were kept identical as reported previously [37].

#### 4.4.4. MS Raw Data Analysis

The Analyst QS 2.0 software (Applied Biosystems) was used for spectral data acquisition. ProteinPilot Software 3.0, Revision Number: 114,732 (Applied Biosystems) was used for the peak list generation, protein identification, and quantification against the concatenated target-decoy Uniprot human database (191,242 sequences, downloaded on 10 March 2012). The protein ratios were calculated from the peptide-level iTRAQ ratios of confidently detected unique peptides after averaging them. The false discovery rates (FDR) of peptide and protein identification were set to be less than 1% (FDR = 2.0 × decoy_hits/total_hits) [19]. Details of the analysis strategy have been described in SI Methods section.

### 4.5. GO Analysis

The GO analysis was performed using Automated Bioinformatics Extractor (ABE) (http://helixweb.nih.gov/ESBL/ABE/, accessed on 25 September 2021). Gene symbols were used as input to retrieve GO terms from three categories; cellular component, biological process, and molecular function. Chi-square (χ2) test of independence was performed to examine the association between regulated proteins and various annotation terms that were obtained through ABE [46]. Significance was considered when *p*-values were <0.05.

### 4.6. Statistical Analyses

All statistical analyses were performed using SPSS 23.0 for Windows software (SPSS Inc., Chicago, IL, USA). One-way ANOVA followed by post hoc Tukey test was used for scale variables such as age. Nonparametric Kruskal–Wallis H test was used for comparing ordinal variables such as demographic characteristics and baseline risk factors. Statistical significance was accepted at *p* < 0.05. The one-sample Kolmogorov–Smirnov goodness-of-fit test was used to check if the log-transformed iTRAQ ratios of MEVs (n = 573) or SEV (n = 229) are normally distributed. The iTRAQ ratios of NAO, RVE, and CD groups did not follow a normal distribution. Bivariate correlations involving continuous, non-normal data (log_2_ (NAO/HC), log_2_ (RVE/HC), log_2_ (CD/HC)) were calculated by nonparametric Spearman’s rank correlation. Statistical significance was accepted at *p* < 0.01 (2-tailed). Correlation coefficients (Spearman’s rho, ρ) were categorized as follows: ρ < 0.20, no correlation; ρ = 0.20–0.40, weak; ρ = 0.40–0.70, moderate; ρ > 0.70, strong.

## 5. Conclusions

We present for the first time a comparative quantitative proteomics dataset of MEVs isolated from the convalescent plasma of LACI patients grouped based on long-term adverse outcome. The major elements of the MEV dataset (573 proteins) are distinct from the SEV dataset and are the largest plasma EV proteome of ischemic stroke to the best of our knowledge. We use a combination of guided data filtering, correlation analysis, GO-coupled enrichment analysis, and manual curation of relevant literature to propose a list of potential prognostic biomarkers for validation on independent and comparable cohorts of LACI patients. A majority of the candidate biomarkers are novel in the context of LACI and are not known to be associated with commonly targeted biological processes such as coagulation, fibrinolysis, endothelial dysfunction, or inflammation. Instead, the deregulated candidates are linked to less-explored biological processes such as endo-lysosomal trafficking, glucose transport, iron homeostasis, and formation of transitional endoplasmic reticulum. The data may help in better understanding the underlying pathology of LACI and associated cSVD. Once validated, some of these biomarkers individually or as a panel can complement existing imaging modalities to predict prognosis in LACI patients. We have created a web resource to facilitate data access, data sharing, meta-analysis, and rational integration of multi-omics data.

## 6. Patents

A part of the work has been used to obtain a US patent (International Publication Number: WO 2015/038065 A1) on ‘‘Plasma Microvesicle Biomarkers for Ischemic Stroke’’.

## Figures and Tables

**Figure 1 ijms-23-11670-f001:**
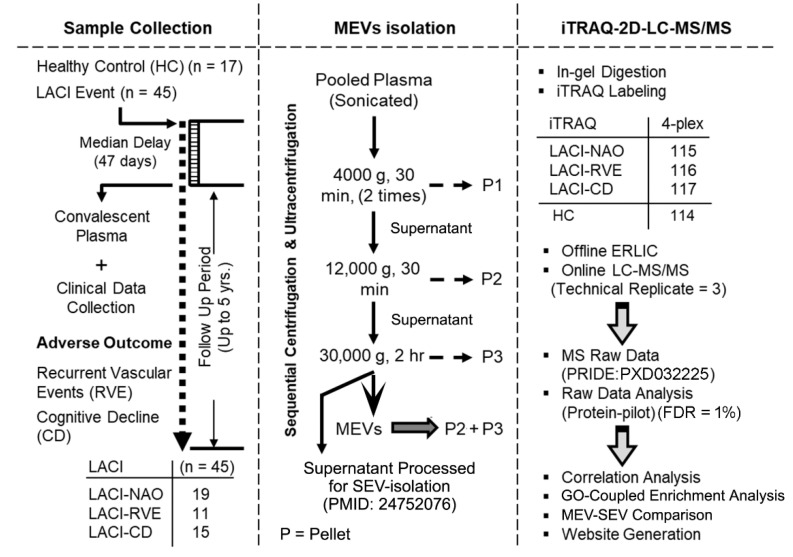
Schematic presentation of the experimental design. The LACI patients are grouped based on adverse outcome (RVE or CD) or no adverse outcome (NAO) when followed up for up to 5 years.

**Figure 2 ijms-23-11670-f002:**
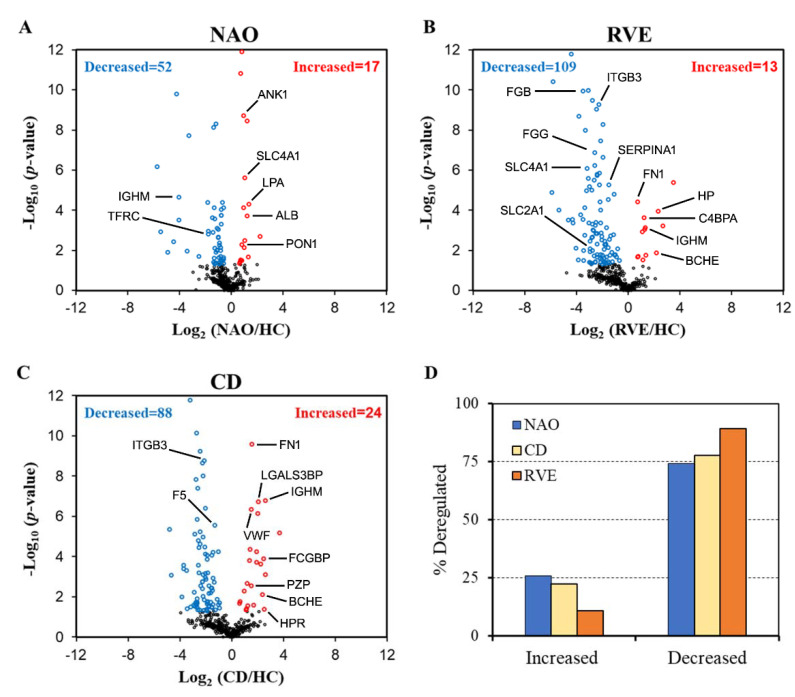
Volcano plot of quantified proteins in three different LACI groups: (**A**) NAO, (**B**) RVE, and (**C**) CD, when compared to HC. The iTRAQ ratios obtained from database searching in ProteinPilot software were log-transformed and plotted against the negative log of the *p*-value for respective ratios. Blue dots indicate decreased and red dots indicate increased proteins satisfying the two filtering criteria; *p* < 0.05 and fold change = 1.5 (|Log_2_(ratio)|> 0.58 and (−log_10_(*p*-value)) > 1.30). The black dots represent unchanged proteins. Some of the shortlisted proteins from Table 1 are labeled. The plots are presented with an axis value of −12.0 to +12.0 for better visualization. (**D**) Bar chart showing the percentage of proteins increased and decreased in three different LACI groups (NAO, RVE, and CD). NAO, no adverse outcome; RVE, recurrent vascular event; CD, cognitive decline; HC, healthy control.

**Figure 3 ijms-23-11670-f003:**
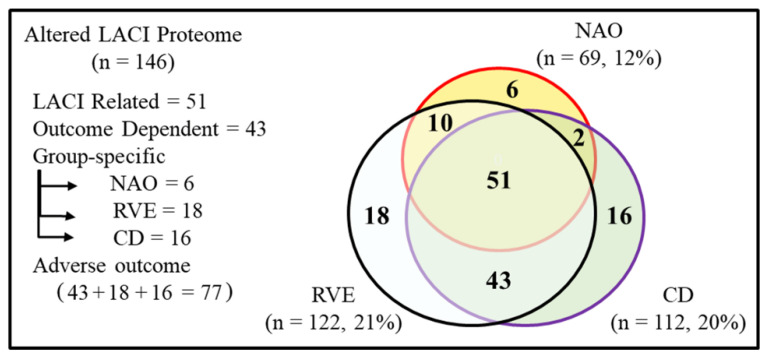
Venn diagram comparing altered proteins quantified by iTRAQ experiment in different LACI groups. A total of 146 out of 573 quantified proteins were used for this analysis. NAO, no adverse outcome; RVE, recurrent vascular event; CD, cognitive decline.

**Figure 4 ijms-23-11670-f004:**
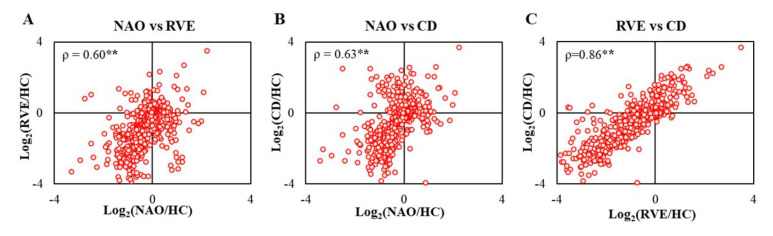
Proteome-wide Spearman rank correlation analysis in between three groups of LACI patients; (**A**) NAO vs. RVE, (**B**) NAO vs. CD, (**C**) RVE vs. CD. Log ratios of all confidently quantified proteins (n = 573) compared to HC (Log_2_(NAO/HC), Log_2_(RVE/HC), Log_2_(CD/HC)) were used for this analysis. The plots were presented with an axis value of −4.0 to +4.0 for better visualization. ρ, Spearman rank correlation; ** *p* < 0.01 (two-tailed).

**Figure 5 ijms-23-11670-f005:**
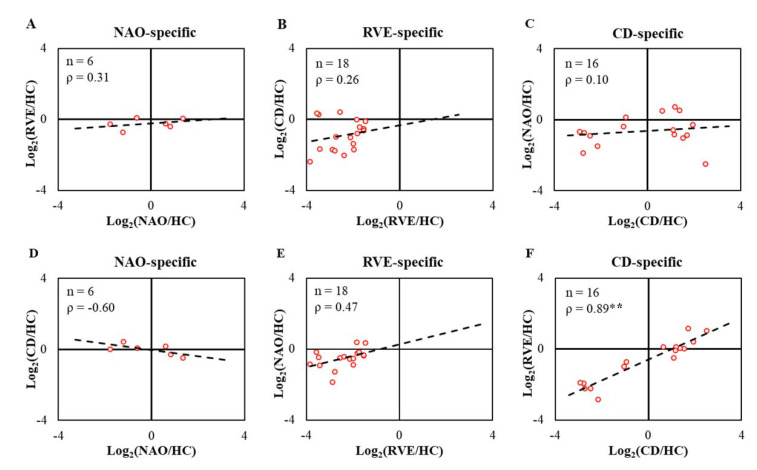
Spearman rank correlation analysis in between significantly altered proteins specific to three LACI groups (NAO, RVE, and CD). (**A**,**D**) Log ratios of proteins altered only in NAO compared to HC (NAO-specific, n = 6, x-axis) were correlated with RVE (RVE/HC) and CD (CD/HC) groups. A similar analysis was performed for proteins changed in RVE (RVE-specific, n = 18, (**B**,**E**)) and CD (CD-specific, n = 16, (**C**,**F**)) groups. A linear trendline was added for easy visualization of the comparative pattern of regulation. ρ, Spearman rank correlation; ** *p* < 0.01 (two-tailed).

**Figure 6 ijms-23-11670-f006:**
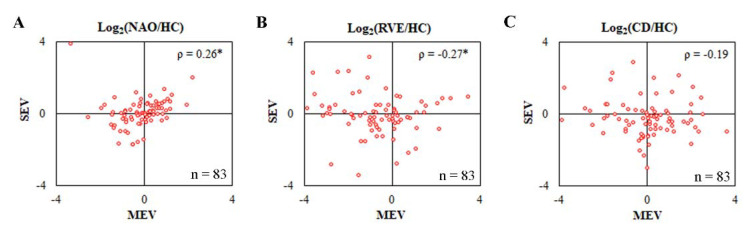
Spearman rank correlation analysis of commonly quantified proteins in MEV and SEV proteomics datasets for three LACI groups; (**A**) NAO, (**B**) RVE, and (**C**) CD. Log ratios of proteins in different outcome groups compared to HC were plotted between MEV and SEV fractions. The plots were presented with an axis value of −4.0 to +4.0 for better visualization. ρ, Spearman rank correlation; n = 83, * *p* < 0.05 (two-tailed).

**Table 1 ijms-23-11670-t001:** Deregulated plasma MEV protein shortlist in LACI patients.

Gene Symbol	Accession	Protein Name	Identification Parameters			Quantitation Ratios ^3^		
			Protein Score ^1^	%Cov(95)	Peptides (95%) ^2^	Log_2_(NAO/HC)	Log_2_(RVE/HC)	Log_2_(CD/HC)
FLII	Q13045	Protein flightless-1 homolog	4.6	1.9	2	−0.58	**−2.11**	**−2.86**
DYNC1H1	Q14204	Cytoplasmic dynein 1 heavy chain 1	6.9	0.4	2	−0.40	−0.98	**−1.04**
AP2B1	P63010-2	Isoform 2 of AP-2 complex subunit beta	7.1	2.7	3	−0.57	**−2.13**	−1.01
BCHE	P06276	Cholinesterase	7.2	7.0	3	−0.12	**2.18**	**2.39**
UBA52	P62987	Ubiquitin-60S ribosomal protein L40	7.4	29.7	3	−0.36	**−1.54**	−0.53
MYLK	Q15746-5	Isoform 4 of myosin light chain kinase, smooth muscle	8.0	2.2	4	−0.47	**−1.67**	**−1.77**
CANX	P27824-2	Isoform 2 of calnexin	8.3	8.0	4	−0.44	**−2.39**	−2.03
KLKB1	H0YAC1	Plasma kallikrein (fragment)	8.4	6.7	4	−0.58	−0.50	**1.09**
KIF2A	O00139-4	Isoform 4 of kinesin-like protein KIF2A	8.4	6.5	4	−0.53	**−1.99**	−1.38
SLC2A1	P11166	Solute carrier family 2, facilitated glucose transporter member 1	10.6	13.2	6	1.04	**−2.75**	−1.00
DIAPH1	A0A0G2JH68	Protein diaphanous homolog 1	10.9	5.0	5	−0.33	**−2.10**	**−1.74**
IQGAP2	Q13576	Ras GTPase-activating-like protein IQGAP2	10.9	1.8	3	0.25	**−1.10**	**−2.02**
PIGR	P01833	Polymeric immunoglobulin receptor	11.3	10.9	6	−0.88	1.14	**1.70**
PECAM1	P16284-6	Isoform Delta15 of platelet endothelial cell adhesion molecule	11.6	10.0	5	−0.45	**−1.21**	**−1.61**
GANAB	Q14697	Neutral alpha-glucosidase AB	11.7	9.7	6	−0.53	**−1.25**	**−1.50**
PROS1	P07225	Vitamin K-dependent protein S	11.9	10.8	6	−0.13	**1.16**	**1.24**
HPR	A0A0A0MRD9	Haptoglobin-related protein	12.3	55.5	39	−2.50	1.04	**2.50**
HSP90B1	P14625	Endoplasmin	13.4	9.2	6	−0.68	**−2.68**	**−1.59**
ADD1	P35611-3	Isoform 3 of alpha-adducin	13.4	9.8	6	0.39	**−1.85**	0.00
FLOT2	E7EMK3	Flotillin-2	14.2	18.0	7	−0.35	**−1.54**	−0.62
APOH	P02749	Beta-2-glycoprotein 1	14.5	28.1	12	−0.27	**−2.76**	**−1.75**
VASP	P50552	Vasodilator-stimulated phosphoprotein	15.0	19.7	7	−0.64	**−3.36**	**−1.42**
VTN	P04004	Vitronectin	17.5	20.9	13	−0.23	**−1.82**	−0.78
CAT	P04040	Catalase	17.6	22.8	8	**0.61**	−0.23	0.16
ATP5A1	P25705	ATP synthase subunit alpha, mitochondrial	18.3	21.2	9	−0.31	**−1.81**	**−1.90**
PZP	P20742	Pregnancy zone protein	18.6	16.8	126	−1.04	0.01	**1.53**
FCGBP	Q9Y6R7	IgGFc-binding protein	19.0	3.1	9	**−1.37**	**1.37**	**2.48**
HPX	P02790	Hemopexin	19.3	21.9	10	**0.82**	**−0.90**	0.11
TFRC	P02786	Transferrin receptor protein 1	19.5	14.2	9	**−1.77**	−0.27	0.01
PON1	P27169	Serum paraoxonase/arylesterase 1	20.0	39.7	11	**0.82**	−0.39	−0.29
KNG1	P01042-2	Isoform LMW of kininogen-1	23.1	28.6	12	**0.69**	**−1.02**	0.24
APOL1	O14791	Apolipoprotein L1	25.3	34.7	14	**−1.33**	0.25	**1.91**
CD5L	O43866	CD5 antigen-like	25.6	41.2	16	**−0.69**	**0.74**	**1.99**
STOM	P27105	Erythrocyte band 7 integral membrane protein	27.0	50.0	20	−0.07	**−0.82**	**−1.34**
GP5	P40197	Platelet glycoprotein V	27.7	33.6	17	−0.45	**−2.15**	**−1.90**
C4BPA	P04003	C4b-binding protein alpha chain	28.1	28.3	19	−0.23	**1.22**	**1.42**
EPB42	P16452	Erythrocyte membrane protein band 4.2	29.5	19.0	16	**0.96**	**−1.53**	−0.72
MSN	P26038	Moesin	29.6	28.4	17	−0.17	**−1.73**	−0.44
ITGB3	P05106	Integrin beta-3	32.8	22.8	21	−0.56	**−2.26**	**−2.13**
LGALS3BP	Q08380	Galectin-3-binding protein	35.9	34.4	23	**−1.12**	**1.10**	**2.03**
APOA1	P02647	Apolipoprotein A-I	39.2	56.9	23	−0.80	**−1.40**	**0.96**
EPB41	P11171-2	Isoform 2 of protein 4.1	39.8	30.6	21	**0.98**	**−2.48**	−0.60
APOE	P02649	Apolipoprotein E	40.1	68.4	25	0.13	−0.72	**−0.94**
VCP	P55072	Transitional endoplasmic reticulum ATPase	43.9	36.6	21	**−0.61**	0.09	0.08
SERPINA1	P01009	Alpha-1-antitrypsin	45.2	55.5	32	0.33	**−1.47**	−0.09
LPA	P08519	Apolipoprotein(a)	46.5	29.2	38	**1.36**	0.05	−0.48
F5	A0A0A0MRJ7	Coagulation factor V	47.3	11.9	22	−0.29	**−2.13**	**−1.29**
FCN3	O75636	Ficolin-3	48.3	64.9	87	0.27	**−1.00**	**−0.93**
VWF	P04275	von Willebrand factor	48.7	9.7	28	−0.43	**−1.09**	**1.49**
IGHA1	P01876	Immunoglobulin heavy constant alpha 1	52.9	56.9	68	−0.85	−0.07	**1.16**
IGKC	P01834	Immunoglobulin kappa constant	54.5	91.6	123	−0.28	0.40	**1.94**
FGG	P02679	Fibrinogen gamma chain	69.2	61.6	99	−0.49	**−2.56**	0.43
HP	P00738	Haptoglobin	73.8	64.5	92	0.29	**2.33**	**2.22**
TF	P02787	Serotransferrin	75.2	52.6	57	**0.58**	**−1.95**	0.07
SLC4A1	P02730	Band 3 anion transport protein	98.1	41.8	99	**1.05**	**−3.16**	**−1.05**
FN1	P02751-15	Isoform 15 of fibronectin	101.6	27.5	67	**−0.85**	**0.76**	**1.54**
FGA	P02671	Fibrinogen alpha chain	105.5	42.8	152	−0.19	**−3.56**	0.33
FGB	P02675	Fibrinogen beta chain	110.1	81.9	118	−0.48	**−3.48**	0.28
ANK1	P16157-14	Isoform Er13 of ankyrin-1	145.5	45.4	110	**0.94**	**−2.25**	−0.41
IGHM	P01871	Immunoglobulin heavy constant mu	192.7	69.1	390	**−4.07**	**1.30**	**2.62**
SPTB	P11277-2	Isoform 2 of spectrin beta chain, erythrocytic	200.6	50.1	129	**0.80**	**−3.20**	−0.53
SPTA1	P02549	Spectrin alpha chain, erythrocytic 1	230.4	57.0	138	**0.73**	**−2.96**	**−0.36**
ALB	P02768	Serum albumin	258.4	85.5	399	**1.21**	**−2.75**	0.05

^1^ The proteins are sorted by “protein score” to show the less abundant candidates on the top. ^2^ The reported numbers are the number of unique peptides identified with at least 95% confidence. ^3^ The log2 ratios with a significant *p*-value (<0.05) are indicated in **bold**.

## Data Availability

The mass spectrometry generated raw data files have been deposited to the ProteomeXchange Consortium (http://proteomecentral.proteomexchange.org/cgi/GetDataset, accessed on 19 August 2022) via the PRIDE partner repository with the dataset identifier PXD032225. The analyzed, classified, and curated data are available at https://yenepoya.res.in/database/LTN_Datta_Lab/LACI_MEV_Proteomics/index.html (accessed on 19 August 2022).

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
