# Peer review of "Quantitative Proteomics of Medium-Sized Extracellular Vesicle-Enriched Plasma of Lacunar Infarction for the Discovery of Prognostic Biomarkers"

_ijms, 2022, doi:10.3390/ijms231911670_

Round 1

Reviewer 1 Report

The manuscript authored by Datta Arnab et al. reports the isolation and proteome analysis of medium-sized extracellular vesicles from LACI plasma patients. The topic is interesting and the article well written. A strong point in their manuscript is that their purified fractions are first characterized according to published guidelines. Also they make some comments about their proteomic data and the correlation between RVE and CD is quite interesting. However the manuscript lacks some details about their methodology and there are also some other aspects which are ambiguous or some results are overrepresented. These are detailed below:

Major:

11.      Please include relevant graphs displaying quantified proteins which are specific to MEV in comparison to SEV (such as but not limited to calnexin, endoplasmin etc). These should denote the number of quantified peptides, the co-isolation level (since iTRAQ is used) and error bars displaying the variability for all measured samples.

22.      P15, L479: The authors report that: ‘The plasma samples were pooled group-wise before processing.’ Is this referring to the iTRAQ? Please clearly describe in the manuscript text how these samples were pooled. How many samples were analyzed in the end on the instrument (technical, analytical biological replicates)?

33.      The authors mention data deposition on PRIDE. However the data cannot be accessed using only the accession number provided. The authors should also provide the username and password in the manuscript content required to access the data. 

44.      For single identification proteins annotated MS/MS spectra should be provided either as supplemental material (PDFs or image files) or deposited in a publicly accessible repository (for example MS-Viewer: https://msviewer.ucsf.edu/prospector/cgi-bin/msform.cgi?form=msviewer ).

55.      Please add PCA plot or correlation heat map between replicates to demonstrate that indeed the dataset can distinguish at the molecular level between the analyzed groups (e.g. NAO, RVE, CD etc.).

66.      Figure 2: please mention in the figure legend the statistical test used and the corresponding correction to calculate the reported p-values. Also, please add in the supplemental histogram distributions of log2 ratios.

77.      P7, Section 2.4: The authors compared the distribution of some GO terms between altered and unaltered MEV proteins. In my opinion this does not clearly demonstrates that MEVs offers advantages when searching for plasma biomarkers. The comparison should have been made with whole plasma, for which the author do not provide any data. This is probably because the frequency of the GO selected terms would probably be higher in MEV proteome, but, even more this does not assume that all expected plasma biomarkers are related to those terms. In my opinion this comparison is not convincing and just counting the terms with ‘vesicle’ or any other specific GO term in altered and unaltered MEV proteome group does not provide any solid evidence for their claims. What GO level of redundancy did the authors used? How many of the proteins were annotated in general? In my opinion this section does not bring any added value to the manuscript and could be removed or at least could go to the supplemental. If the authors wish to keep this paragraph they should better argue why the different term distribution between altered and unaltered MEV proteins brings any added value.

88.      P8, Section 2.5: There is a little bit of confusion regarding the terms used by the authors in the manuscript. The authors use the term ‘plasma proteome’ in some instances when referring to their dataset. However we should note that the authors analyzed only the MEV plasma proteome and not the plasma proteome, thus their results cannot be extended to the full plasma of the patients since I did not find any section in which the authors report the full plasma proteome, only the MEV proteome fraction. Please clearly denote throughout the manuscript if at any step the full plasma proteome was analyzed or only the MEV fraction. If this is the case then please adjust the text accordingly, not to create any confusion for the reader.

Minor:

11.      Figure 1: Please describe also in the figure legend: LACI-NAO, LACI-RVE and LACI-CD.

22.      P4, L135: What do the author considers intermediate purity? Please describe in the text.

33.      P15, L500: Why do the authors choose to use MMTS for alkylation? Unlike typical alkylation reagents this is reversible by reducing buffers (DTT or TCEP). Did they re-assured to remove all excess TCEP before alkylating?

44.      P15, L503: The authors use 2% ACN? Why? What buffers did they used for peptide extraction?

55.      P15, L508: Did the authors pooled the fractions collected following ERLIC fractionation? If yes based on what criteria?

66.      P15, L527: Please add the version of the UniProt database used, not only the no. of sequences: when it was downloaded?

77.      P16, L528-530: Please provide PSM FDR, peptide FDR and protein FDR used. Also, does the FDR reported is obtained using the target-decoy strategy or an alternative one. Please explain in the text.

88.      Please also report the number of unique peptides not only the peptides no. in the supplemental tables.

99.      The Methods section reports data filtering to 1%FDR, but the supplemental tables report Peptides (95%)? Why? Is it 1% or 5% FDR? Please describe or adjust accordingly.

110.  Figure 2: Do the authors report mean or median of multiple measurements regarding log2 ratios? Please mention in the text. Also, please add labels to points from the volcano plots denoting most relevant proteins.

111.  P8, Sections 2.4 and 2.5: Actually I find that a more interesting comparison regarding GO terms would have been between MEV fraction proteome and the plasma proteome during Section 2.1.

112.  Please describe in the methods section how the missing values were handled.

113.  P9, Table 1: Please report the number of unique peptides for each entry. Also do the identification parameters refer to the aggregated dataset?

114.  P11, L309-312: The authors cannot claim that unreported proteins are novel biomarkers. This has to be further validated. Thus please change the text accordingly.

115.  P10, L325-327: The authors state that MEV results in better proteome coverage compared with SEV. However, how do they know the real proteome dimension of both to claim the increased coverage? Just based on the number of identified proteins?

Author Response

Please see the attachment - Point by point answer to queries raised by Reviewer 1

Reviewer 2 Report

Lacunar infarction (LACI), a subtype of acute ischemic stroke, has an increased prevalence among south Asians when compared to whites and is etiologically the most common subtype in the Japanese population. LACI is a vasculature-linked disorder that occurs due to the thickening of vascular wall and subsequent narrowing of deep penetrating cerebral arteries. This causes endothelial dysfunction and an impaired autoregulation of cerebral blood flow. Silent or symptomatic LACI can also be associated with systemic endothelial dysfunction that is present in vascular beds in the periphery (kidney or brachial artery). LACI has poor mid- to long-term prognosis. This study aimed to discover blood-based biomarkers for LACI as a complementary prognostic tool by proteomic analysis of a pool of plasma collected from 45 patients with lacunar stroke and controls. The major result is that out of 573 (FDR <1%) quantified proteins, 146 showed significant changes in at least one LACI group when compared to matched healthy controls. 

Comments

The study is welcome and of interest to clinicians. However, several points have to be clarified/improved:

1)     INTRODUCTION: plasma EV have been also used for therapeutical purposes (see, doi: 10.1161/STROKEAHA.119.028012)

2)     ) METHODS: a major drawback is that the analysis was done on a relatively small pool. In this case a validation using individual plasma samples has to be done. Have been comorbidities taken into account at patients’ recruitment?

3)     DISCUSSION shall start with the major conclusion of the study. A recent study on predictive biomarkers of ischemic stroke has been published. A final conclusion is also lacking.

Author Response

Please see the attachment - Point by point answer to queries raised by Reviewer 2

Reviewer 3 Report

In this article, Datta et al. use quantitative proteomics of MEV-enriched plasma of LACI patients (with no adverse outcome, with recurrent vascular event only, or with cognitive decline only), and describe a number of proteins that might possess a complementary prognostic value of such three conditions. The paper is well written and structured and, using a very similar approach, shows new data from plasma samples taken in a study done a few years ago (from 1999 to 2006), from which the authors already published a couple of papers in mid 10s. Since the discovery of consistent stroke biomarkers, in its different aspects and subtypes, is still a pending issue in the field, the research efforts done in this regard the have much merit.

I am concerned about several issues.

1 Direct comparisons made between MEV-enriched plasma proteome of the present paper and SEV-enriched plasma proteome published 7-8 years earlier.

2 The use of aspirin with or without dipyridamole in LACI patients but not in controls.

3 The stabiliby of some proteinaceous species in plasma and EV in plasma after such a long storage time.

4 Also, had the plasma samples used in this study been previously thawed-frozen?

5 It is not clear to me when the plasma samples used in the present study were taken after the LACI event and if it was the same for all subjects. This might be an important issue as some putative blood-based stroke biomarkers have been reported to show a temporal pattern.

6 Although it is common practice in proteomic studies to pool plasma obtained from subjects of the same experimental group, this is not devoid of possible confounding artifacts, e.g. the presence of an outlier with very high levels of one or more proteins. Further WB/ELISA of the most promising candidates would have helped in this regard.

7 IMO, it would have been highly informative to include a group of patients with recurrent vascular event plus cognitive decline.

8 Plasma samples from control subjects were obtained years later than those of LACI patients. Also, was age in any of the LACI groups statistically different from that of controls?

Minor issues

9 It would be convenient to locate geographycally and reference sentence in lines 38-39.

10 In the context of this biomarker discovery study, why plasma is referred to as ‘convalescent plasma’?

11 Lines 94-95: SEV-proteome?

Author Response

Please see the attachment - Point by point answer to queries raised by Reviewer 3

Round 2

Reviewer 1 Report

The author have addressed most of my concerns and the manuscript can be considered for publication.

Reviewer 2 Report

The authors have successfully addressed my observations. The manuscript can be published in its present form.

Reviewer 3 Report

The authors have adequately addressed my concerns and changed the text in the MS accordingly.